# The Utilization of Optically Induced Dielectrophoresis (ODEP)-Based Cell Manipulation in a Microfluidic System for the Purification and Sorting of Circulating Tumor Cells (CTCs) with Different Sizes

**DOI:** 10.3390/mi14122170

**Published:** 2023-11-29

**Authors:** Po-Yu Chu, Thi Ngoc Anh Nguyen, Ai-Yun Wu, Po-Shuan Huang, Kai-Lin Huang, Chia-Jung Liao, Chia-Hsun Hsieh, Min-Hsien Wu

**Affiliations:** 1Graduate Institute of Biomedical Engineering, Chang Gung University, Taoyuan City 33302, Taiwan; d000018394@cgu.edu.tw (P.-Y.C.); d1131003@cgu.edu.tw (T.N.A.N.); m1031001@cgu.edu.tw (A.-Y.W.); leo_6813@cgu.edu.tw (P.-S.H.); m1031009@cgu.edu.tw (K.-L.H.); 2Department of Biomedical Sciences, College of Medicine, Chang-Gung University, Taoyuan 333, Taiwan; l329735@cgu.edu.tw; 3Division of Hematology-Oncology, Department of Internal Medicine, New Taipei City Municipal Tucheng Hospital, New Taipei City 23652, Taiwan; wisdom5000@cgmh.org.tw; 4Division of Hematology-Oncology, Department of Internal Medicine, Chang Gung Memorial Hospital at Linkou, Taoyuan City 33302, Taiwan

**Keywords:** optically induced dielectrophoresis (ODEP), microfluidic technology, circulating tumor cells (CTCs), cell isolation, cell sorting

## Abstract

The analysis of circulating tumor cells (CTCs) at the molecular level holds great promise for several clinical applications. For this goal, the harvest of high-purity, size-sorted CTCs with different subtypes from a blood sample are important. For this purpose, a two-step CTC isolation protocol was proposed, by which the immunomagnetic beads-based cell separation was first utilized to remove the majority of blood cells. After that, an optically induced dielectrophoresis (ODEP) microfluidic system was developed to (1) purify the CTCs from the remaining magnetic microbeads-bound blood cells and to (2) sort and separate the CTCs with different sizes. In this study, the ODEP microfluidic system was designed and fabricated. Moreover, its optimum operation conditions and performance were explored. The results exhibited that the presented technique was able to purify and sort the cancer cells with two different sizes from a tested cell suspension in a high-purity (93.5% and 90.1% for the OECM 1 and HA22T cancer cells, respectively) manner. Overall, this study presented a technique for the purification and sorting of cancer cells with different sizes. Apart from this application, the technique is also useful for other applications in which the high-purity and label-free purification and sorting of cells with different sizes is required.

## 1. Introduction

Cancer has been a primary threat to global health in recent decades. It is well known that the majority of cancer deaths are due to cancer metastasis [1,2]. Circulating tumor cells (CTCs) are the rare cancer cells existing in the peripheral blood of cancer patients, which were first described in 1869 [3]. Several studies have well demonstrated that the presence of CTCs in blood circulation is relevant to cancer metastasis [4]. Therefore, the analysis of CTCs holds great promise for several clinical utilizations (e.g., cancer detection [4,5], prognosis evaluation of cancer [6,7], therapeutic response evaluation of cancer [8,9,10], or monitoring of cancer metastasis [11,12]). Compared with the clinical applications of CTCs simply based on the quantification of their cell number [6,11], the analysis of CTCs at molecular level (e.g., gene expression analysis) could pave a more promising route to comprehensively understand CTC heterogeneity [13,14], cancer progression [15,16], or mechanisms of therapeutic resistance [17,18]. To achieve this goal, the harvest of high-purity and all possible CTCs with different subtypes from the blood samples of cancer patients is crucially important.

However, CTCs are naturally rare in a blood sample (e.g., approximately 1 CTC per 10^5^–10^7^ leukocytes [19]), making them technically challenging to isolate and purify. In terms of the CTC isolation and purification schemes, they can be generally categorized into cell biophysical- and immunoaffinity-based mechanisms [20]. For the former, the physical (e.g., size or density) differences between CTCs and the background blood cells [21] are mainly used to separate and then purify CTCs from a blood sample. This strategy is generally regarded to be simple and low cost to operate. However, its purity and specificity for target cell isolation are commonly inferior to the immunoaffinity-based strategy [22]. For the immunoaffinity-based methods, their working principles are based on the use of specific antibodies to selectively bind to the surface antigens of cells to distinguish the target CTCs from the other blood cells. CTCs normally express specific cellular surface antigens (e.g., EpCAM and CKs) that set them apart from the other blood cells (i.e., positive immunoselection). Alternatively, the CTC isolation and purification can be achieved by specifically removing the majority of blood cells, leaving the cell population that do not express any blood cell-related antigens for the subsequent collection (i.e., negative immunoselection) [23].

Due to the heterogeneity issue of CTCs regarding their surface antigens, the utilization of the positive immunoselection-based strategy might only be able to harvest some possible CTCs with different subtypes [24]. Conversely, the negative immunoselection-based CTC isolation and purification techniques are believed to obtain all possible and label-free CTCs from a blood sample [25]. Due to this technical advantage, the subsequent CTC analysis could produce more comprehensive information, revealing more real situations of cancer status. In practice, briefly, only blood cells (e.g., leukocytes) are targeted for removal via a standard immunomagnetic microbeads-based cell separation and isolation technique [25]. Nevertheless, the CTC isolation and purification based on this strategy generally comes across the technical problem of low CTC purity [25]. In this situation, the contamination of leukocytes in the obtained cell sample could, therefore, complicate the subsequent CTC analysis. Apart from the abovementioned issues, it was also reported that the CTCs in blood samples have a diverse array of sizes (e.g., diameter: 12∼30 μm) [26,27], reflecting that cancer cells maintain their high phenotypic plasticity in the bloodstream. Moreover, by analyzing the morphological features of CTCs in different cancer types, the results reveal that the CTCs isolated from hepatocellular and breast cancer patients exhibit larger cell size compared with those obtained from the other types of cancers [28,29,30]. This finding indicates that CTCs may display unique morphological features (e.g., size) depending on the specific origin of the tumor. As a result, the initial sorting and separation of CTCs based on their size differences during the CTC isolation and purification process might facilitate the following CTC analytical work.

Moreover, with the advances in microfluidic technology, the microfluidic systems integrated with various techniques (e.g., fluidic control- [31], acoustophoresis- [32], dielectrophoresis (DEP)- [33], or optically induced dielectrophoresis (ODEP)- [34] based mechanism) for cell manipulation have been presented for the isolation and purification of biological cells. Due to the feature of miniaturization, overall, these microfluidic systems have been proved to have superior cell isolation and purification performance over conventional cell isolation and purification schemes [31]. Among the cell manipulation mechanisms, ODEP-based techniques are particularly suitable for biologists to use owing to their ease of fabrication and operation. The microparticle manipulation using ODEP was first presented in 2005 [35]. Its working principle is well discussed elsewhere and briefly described herein. An electric voltage is applied within the solution layer of an ODEP system to generate a uniform electric field. This in turn causes the dielectric microparticles (e.g., cells) within the solution to be electrically polarized. When the photoconductive substrate of the ODEP system is projected with light, the illuminated light can cause the electrical impedance at the light-projected area to decrease. This phenomenon in turn causes the applied electric voltage to decline across the solution layer within the light-illuminated region. Overall, the abovementioned phenomenon creates a locally non-uniform electric field within an ODEP system. Based on these facts, the interaction between the non-uniform electric field created and an electrically polarized microparticle is used to manipulate the microparticles. In terms of operation, a scientist can simply control the dynamic optical images projected on an ODEP system to manipulate the microparticles in a manageable manner.

To harvest high-purity and all possible CTCs with different sizes from a blood sample, a two-step CTC isolation and purification protocol was proposed in this study. For the first step, the negative selection-based CTC isolation scheme based on the immunomagnetic beads-based cell separation technique was adopted to remove 99% of blood cells [36]. In the subsequent step, the combination of ODEP-based cell manipulation and the laminar flow phenomenon in the presented microfluidic system was utilized to (1) purify the label-free CTCs from the remaining magnetic microbeads-bound blood cells, and to (2) further sort and separate the CTCs with different sizes. In this study, research was mainly focused on the second step. For the ODEP microfluidic system, its working principles are mainly based on the phenomena described in the following. First, the ODEP force acting on the cells with varied sizes is different [37]. Second, previous studies have demonstrated that the ODEP force generated on a cell is influenced by the intensity (or color brightness ratio) of illuminated light [38,39,40]. Based on these facts, two dynamic light image arrays were designed in the microchannel (i.e., the defined zone for cell separation) to first gather all cells to one side of the microchannel and then to sort and separate the magnetic microbeads-bound blood cells and the cancer cells with different sizes, respectively. Moreover, due to the laminar flow pattern in a microchannel, the separated cells were then transported in a partitioned manner for the subsequent cell collections via two static light bar arrays.

In this work, the ODEP microfluidic system was designed and fabricated. Moreover, the ODEP operating conditions (i.e., the moving velocities and color brightness of the light bars used in the two dynamic light image arrays, and the color brightness of the light bars used in the two static light bar arrays) for effective purification and sorting of cancer cells with two different sizes were explored experimentally. For performance evaluation purposes, two cancer cell lines (OECM1 and HA22T cells) and a Jurkat cell line (used as model leukocytes) were used as the model cells representing the CTCs with different sizes and the remaining blood cells in the processed sample. Finally, the performance of the proposed microfluidic system for the purification and sorting of cancer cells with two different sizes was assessed. The results demonstrated that the ODEP-based cell manipulation scheme was able to effectively purify and sort the cancer cells with two different sizes from a tested cell suspension model in a high-purity (achieved purity for the OECM1 and HA22T cancer cells: 93.5% and 90.1%, respectively) manner. Overall, this study presented an ODEP microfluidic system for the purification and sorting of cancer cells with different sizes. In addition to the application for CTC purification and sorting, the proposed approach is also useful in other applications in which the high-purity and label-free purification and sorting of cells with different sizes are required.

## 2. Materials and Methods

### 2.1. Design, Fabrication, and Experimental Setup of ODEP-Based Microfluidic System for the Purification and Sorting of CTCs with Different Sizes

In this work, an ODEP-based microfluidic chip was designed to (1) purify the CTCs from the magnetic microbeads-bound blood cells and to (2) further sort and isolate the CTCs with different sizes in a continuous manner. The design of the ODEP microfluidic chip is schematically shown in Figure 1a. Briefly, a microchannel (L: 25 mm, W: 1 mm, H: 50 µm) was designed to transport cell suspension samples. In this work, the ODEP-based cell manipulation was conducted within the defined cell separation zone of the microchannel, as indicated in Figure 1a. In the design (Figure 1a), two through-holes (D: 1 mm) for tubing connection were used for the fresh cell suspension sample input and the waste sample output, respectively. The layer-by-layer structure of the ODEP microfluidic chip is illustrated in Figure 1b. Briefly, the microfluidic chip consisted of a top processed polydimethylsiloxane (PDMS) substrate (Layer A), an indium-tin-oxide (ITO) glass (Layer B), a processed double-sided adhesive tape (thickness: 50 µm) with a hollow microchannel (Layer C), and a bottom ITO glass coated with a layer of photoconductive material (Layer D).

The fabrication and experimental setup have been well described previously [36,38]. Briefly, the PDMS tube connector (Layer A; Figure 1b) was made by the CNC (EGX-400, Roland Inc., Hamamatsu, Japan) machining for positive polymethylmethacrylate (PMMA) mold fabrication and the subsequent PDMS (Sylgard^®^ 184, Dow Corning, Midland, MI, USA) replica molding for the tube connector making. For the ITO glass (InnoLux Corporation, ZMI, Taiwan) with two through-holes (Layer B; Figure 1b), the holes were fabricated via mechanical drilling. For Layer C (Figure 1b), the hollow structure of the microchannel was fabricated in the double-sided adhesive tape (L298, Sun-yieh, TYN, Taiwan) simply via a manual punching process using a custom-made metal mold. For Layer D (Figure 1b), an ITO glass was deposited with a photoconductive material (i.e., a 20 nm thick n+ hydrogenated amorphous silicon layer (n^+^ a-Si:H) and a 1 μm thick intrinsic hydrogenated amorphous silicon (a-Si:H) layer) via plasma-enhanced chemical vapor deposition (PECVD).

For the following assembly process, the Layer A was bonded with the Layer B via O_2_ plasma surface treatment. This was followed by the assembly with the Layer D via the Layer C (i.e., a double-sided adhesive tape). In terms of operation, a syringe pump (KDS LEGATO 180, KD Scientific, Holliston, MA, USA) was used to transport the cell suspension sample in the microchannel. For cell manipulation using ODEP, a function generator (AFG-2125, Good Will Instrument Co., Ltd., New Taipei City, NTPC, Taiwan) was used to generate an alternating current (AC) voltage between the two ITO glass substrates (Figure 1b). In the following step, a computer-controlled projector (EB-X05, Epson, Suwa, Japan) was used to illuminate dynamic light images onto the Layer D. In this work, the observation and recording of the cell manipulation process was achieved by using a CCD-equipped microscope (Zoom 160, OPTEM, Medina, OH, USA). The overall experimental setup is schematically illustrated in Figure 1c.

### 2.2. The Working Principle of the ODEP-Based Cell Manipulation Scheme for the Purification and Sorting of CTCs with Different Sizes

The mechanism of the cell manipulation using ODEP is well discussed elsewhere [36,41] and has also been briefly described in the introduction section. The ODEP force generated on a microparticle can be expressed by the Equation (1), also used to describe the DEP force [36,41]:F_DEP_ = 2πr^3^ε_0_ε_m_Re[f_CM_]∇|E|^2^(1)
where r, ε_0_, ε_m_, ∇|E|^2^, and Re[f_CM_] denote the microparticle’s radius, vacuum permittivity, relative permittivity of the surrounding solution, gradient of the exerted electrical voltage squared, and real part of the Clausius–Mossotti factor (f_CM_), respectively. It can be found from Equation (1) that the ODEP force generated on a microparticle is proportional to its radius cubic under the given operation conditions (e.g., the applied electrical voltage or the property of solution) [36,41]. Therefore, one can use ODEP-based cell manipulation to sort and separate the cells with different sizes based on their different ODEP manipulation force [36,41]. Apart from the factors described in Equation (1), the ODEP force generated on a microparticle is also increased with the intensity increase in the illuminated light, which was reported to be proportional to its color brightness ratio according to our previous study [38,39].

Based on the phenomena abovementioned, this study proposed a two-step CTC isolation and purification protocol for the isolation, purification, and sorting of CTCs with different sizes from a blood sample. For the first step, a blood sample is treated with the negative selection-based CTC isolation process utilizing a standard immunomagnetic microbeads-based cell isolation approach, which is well described elsewhere [36], to remove the majority of the blood cells in a blood sample. In the second step, the combination of ODEP cell manipulation and laminar flow phenomenon in the presented ODEP microfluidic system was used to (1) purify all possible CTCs from the remaining magnetic microbead-bound blood cells, and to (2) further sort and separate the CTCs with different sizes. In this study, research mainly focused on the second step. For the development work, two cancer cell lines (OECM1 and HA22T cancer cells; diameter: 20.9 ± 2.6 and 26.6 ± 3.3 μm, respectively) and a Jurkat cell line (used as model leukocytes [36]) were used as the model cells representing the CTCs with different sizes and the remaining blood cells in the processed sample. In the ODEP microfluidic system, briefly, two dynamic light image arrays were designed in the defined cell separation zone, first to gather all cells to one side of the microchannel and then to sort and separate the magnetic microbeads-bound cells and the cancer cells with different sizes, respectively. Furthermore, due to the laminar flow pattern in a microchannel, the separated cells as abovementioned were then transported in a partitioned manner for the subsequent cell collections via two static light bar arrays. The entire operation procedures are illustrated in Figure 2.

### 2.3. The ODEP Operation Conditions for the Purification and Sorting of Cancer Cells with Different Sizes 

In the ODEP microfluidic system, ODEP cell manipulation was utilized to continuously purify and sort the cancer cells from the background magnetic microbeads-bound Jurkat cells, as illustrated in Figure 2. In this work, the electric voltage and frequency were set at 10 Vpp and 3 MHz, respectively, which was previously demonstrated to be a cell-friendly condition for cells [41]. Under the set electric condition, the ODEP manipulation force (i.e., the net force between the ODEP force and friction force producing on a cell) of the cells manipulated (i.e., the Jurkat, magnetic microbeads-bound Jurkat, OECM1, and HA22T cells) was then experimentally evaluated based on the method described previously [36,41]. In a steady state, the ODEP manipulation force acting on a cell is balanced by the viscous drag of fluid. As a result, the hydrodynamic drag force of a moving cell was used to evaluate the net ODEP manipulation force of a cell [36,41]. Stokes’ law (Equation (2)) describes the hydrodynamic drag force (*F*) acting on a spherical particle under continuous flow conditions [36,41]:*F* = 6π*rηv*
(2)
where *r*, *η*, and *v* represent the cell radius, fluid viscosity, and maximum velocity of the cell, respectively. According to Stokes’ law, thus, the ODEP manipulation force acting on the cell investigated can be experimentally assessed through measurements of the maximum velocity of a moving light image that can manipulate the cell [36,41]. Moreover, as described earlier, the ODEP force generated on a cell is influenced by the intensity (or color brightness ratio) of illuminated light, according to our previous study [38,39]. The previous evaluation of the ODEP manipulation force of the manipulated magnetic microbeads-bound Jurkat, OECM1, and HA22T cells was based on the use of the light image with 100% color brightness. In this study, similar evaluations were also carried out using the light images with varied percentages (i.e., 60, 70, 80, 90, and 100%) of color brightness.

### 2.4. Design of the Light Image Arrays in ODEP Microfluidic System for the Purification and Sorting of Cancer Cells with Two Different Sizes

In this study, the combination of ODEP cell manipulation and laminar flow phenomenon in the ODEP microfluidic system was designed to (1) purify the cancer cells (i.e., OECM1 and HA22T cells) from the magnetic microbead-bound Jurkat cells, and to (2) further sort and separate the cancer cells with different sizes. As described in Figure 2, two dynamic light image arrays (i.e., the dynamic light image array I (light bar number: 14; color brightness: 100%; size of light bar: 50 µm (W) and 1316 µm (L); 50 µm (W) and 1459, 1005, and 580 µm (L) for the three bars near the side of the microchannel) and the dynamic light image array II (light bar number: 14; size of light bar: 50 µm (W) and 870 µm (L); color brightness: 100% and 80% as indicated in Figure 2)) were designed in the defined cell separation zone of the microchannel to first gather all cells to one side of the microchannel, and then to sort and separate the magnetic microbeads-bound Jurkat cells and the cancer cells with different sizes, respectively. Moreover, two static light bar arrays (i.e., the static light bar array I (light bar number: 3; color brightness: 80%; size of light bar: 75 µm (W) and 210, 295, and 380 µm (L) for the three bars, respectively) and the static light bar array II (light bar number: 3; color brightness: 100%; size of light bar: 75 µm (W) and 567, 650, and 738 µm (L) for the three bars, respectively)) were designed at the downstream part of the microchannel for the collection of the sorted and separated cancer cells with different sizes, respectively. To achieve this goal, the operation conditions of the two dynamic light image arrays (i.e., the percentage of color brightness and the moving velocity) and the two static light bar arrays (i.e., the percentage of color brightness) were determined based on the previous experimental evaluations, as described in the Section 2.3. The determined operation conditions for the four light image arrays were described and discussed in the Section 3.3. 

### 2.5. Performance Evaluation of the ODEP Microfluidic System for the Purification and Sorting of Cancer Cells with Two Different Sizes

After the dynamic and static light image arrays were designed, their function for the manipulation of the three types of cells as aforementioned was then tested. The function of the dynamic light image array I to manipulate the microbead-bound Jurkat cells to one side of the microchannel (Figure 2b,c) was first experimentally evaluated. In this study, the magnetic microbead-bound Jurkat cell suspension with different concentrations (2.5∼10 × 10^4^ cells 100 μL^−1^) was loaded in the microfluidic chip. The magnetic microbead-bound Jurkat cells transported to the area of the dynamic light image array I was then manipulated using the designed array. The video of the operation process was recorded for quantifying the cells transported to the dynamic light image array I as well as the cells released by the three light bars most near the side of the microchannel (Figure 2c). Based on the data, the removal rate of magnetic microbead-bound Jurkat cells [(the number of cells released by the three light bars most near the side of the microchannel)/(the number of cells entered to the area of dynamic light image array I) × 100%] under different concentrations of cell suspension was then calculated. After understanding the working capacity of the dynamic light image array I to manipulate the magnetic microbead-bound Jurkat cells to one side of the microchannel, the function of the dynamic light image array II to sort and separate the cancer cells with different sizes was experimentally evaluated. Briefly, the magnetic microbead-bound Jurkat, HA22T, and OECM1 cells (concentration: 5 × 10^4^, 5 × 10^3^, and 5 × 10^3^ cells 100 μL^−1^, respectively) were individually loaded into the ODEP microfluidic chip. The loaded cells were then treated with the proposed ODEP manipulation scheme as illustrated in Figure 2. For evaluating the distribution of the cells after ODEP operation, the number of the loaded cells and the number of the cells finally collected by the static light bar arrays I and II were quantified with the aid of video recording.

After the basic function tests as described above, the performance of the proposed ODEP microfluidic system for the purification and sorting of cancer cells with two different sizes was then evaluated. As described earlier, the ODEP microfluidic system was mainly designed to process the blood sample treated with the immunomagnetic beads-based cell separation for removing 99% of blood cells. Therefore, the main purpose of the ODEP microfluidic system was to (1) purify the cancer cells from the magnetic microbead-bound cells, and (2) further sort and separate the cancer cells with different sizes. For evaluating its working performance, the cell suspension tested, containing the magnetic microbead-bound Jurkat, HA22T, and OECM1 cells (ratio: 1:1:5) mimicking a blood sample processed with the immunomagnetic beads-based cell separation process, was prepared and then loaded into the ODEP microfluidic system. After the ODEP manipulation as described in Figure 2, the numbers of the three types of cells collected in the two static light bar arrays were quantified microscopically. For identifying the cell species, HA22T and OECM1 cancer cells were pre-stained with CellTrace™ Calcein Red-Orange (C34851, Invitrogen, Carlsbad, CA, USA) or expressed with GFP (green fluorescent protein), respectively. After that, the purity of the two cancer cells collected in the two static light image arrays was then evaluated.

### 2.6. Statistical Analysis

In this study, the data were presented as the mean ± standard deviation (3 separate experiments). One-way ANOVA was used to evaluate the effect of the experimental factors tested on the outcomes. Tukey’s honestly significant difference (HSD) post hoc test was used to compare differences between the two conditions tested when the null hypothesis of the ANOVA was rejected.

## 3. Results and Discussion

### 3.1. Technical Advantages of the ODEP Microfluidic System for the Purification and Sorting of CTCs with Different Sizes

In terms of application, an ODEP microfluidic system (Figure 1) was developed and used to process further the cell suspension sample obtained from the blood sample treated with the immunomagnetic beads-based cell separation process to specifically remove the majority of blood cells (i.e., the proposed two-step CTC isolation and purification protocol as described earlier). In this work, the proposed ODEP microfluidic system was to (1) purify the CTCs from the remaining magnetic microbeads-bound blood cells, and (2) further sort and separate the CTCs with different sizes (Figure 2). Based on the operations, high-purity, label-free, and all possible CTCs with different size from a blood sample can be harvested, which is both biologically and clinically meaningful for the subsequent medical applications (e.g., prognosis evaluation of cancer [6,7], therapeutic response evaluation of cancer [8,9,10], or monitoring of cancer metastasis [11,12]) or fundamental studies (e.g., mechanisms behind cancer metastasis [15] or anti-cancer drug resistance [17,18]). This technical advantage is beyond what is possible by using other CTC isolation and purification methods by which the CTCs harvested are limited to certain CTC subtypes (e.g., EpCAM-positive CTCs [42]), are labeled with magnetic microbeads [42], or are in a low-purity manner (e.g., biophysical- and negative immunoselection-based CTC enrichment techniques [22,36]).

Among the techniques (e.g., fluidic control- [31], acoustophoresis- [32], DEP- [33], or ODEP- [34] based techniques) available for cell manipulation in microfluidic systems, DEP-based cell or bacteria manipulation is commonly utilized for various applications [33]. Nevertheless, it generally requires a time-consuming, technically demanding, and costly microfabrication to create a special metal electrode array. This technical requirement could therefore restrict its widespread applications. Conversely, the key technical advantage of using ODEP cell manipulation in a microfluidic system is that it can easily and quickly generate or modify an electrode array in a virtual manner via the control of the optical patterns, functioning as a virtual electrode, in an ODEP system. In practice, therefore, a scientist can use a commercial digital projector to display optical images on an ODEP system to manipulate cells in a flexible, manageable, and user-friendly manner through a computer-interfaced control [34]. Overall, this technical feature is particularly suitable for biologists to use owning to their ease of fabrication and operation.

### 3.2. The ODEP Operation Conditions for the Manipulation of Cells

In the ODEP microfluidic system, two dynamic light image arrays were designed in the defined cell separation zone of the microchannel to, first, gather all cells to one side of the microchannel and, then, to sort and separate the magnetic microbeads-bound Jurkat cells and the OECM1 and HA22T cancer cells, as illustrated in Figure 2. Moreover, two static light bar arrays were designed at the downstream part of the microchannel for the collection of the sorted and separated cancer cells with different size, respectively. To achieve this goal, the ODEP operation conditions for the manipulation of the cells (i.e., magnetic microbeads-bound Jurkat, OECM1, and HA22T cells) were first explored. In this study, the electric voltage and frequency for ODEP-based cell manipulation were set at 10 Vpp and 3 MHz, respectively, which was demonstrated to be cell friendly [41]. In this study, the sizes of the model cells used were first evaluated microscopically. The results (Figure 3a) showed that the sizes of Jurkat, OECM1, and HA22T cells were measured to be 13.8 ± 1.7, 20.9 ± 2.6, and 26.6 ± 3.3 μm in diameter, respectively, which were evaluated to have significant difference (*p* < 0.05). After that, the evaluation of their ODEP manipulation force using the indicator of the maximum velocity of a light image that can manipulate the cells [36,41] was carried out. The results (Figure 3b) exhibited that the maximum velocities of a light image that can manipulate the cells (i.e., Jurkat cells and the OECM1 and HA22T cancer cells) were measured to be 105.7 ± 21.2, 138.8 ± 16.5, and 184 ± 20.3 μm s^−1^, respectively. Overall, the measured maximum velocities of a light image that can manipulate the cells increased with the increase in cellular sizes (Figure 3a,b), which were statistically evaluated to have significant differences (*p* < 0.05). This finding is in line with the fact that the ODEP force acting on a cell is proportional to its size, as described in the Equation (1) [36,41]. As discussed previously, a two-step CTC isolation and purification protocol was proposed in this study for harvesting high-purity and all possible CTCs with different sizes. After the first step process as described earlier, the Jurkat cells, the tested model cells representing the leucocytes in this study, should be labeled with magnetic beads. In this situation, the size or electric property of the Jurkat cells might be altered, which could in turn affect their ODEP manipulation force. Nevertheless, the maximum velocities of a light image that can manipulate the Jurkat cells with or without magnetic beads binding were evaluated to have no significant difference (118 ± 21.2 and 105.7 ± 21.2 μm s^−1^, respectively) (*p* > 0.05) (Figure 3b). This result could be partially due to their similar size (i.e., 14.1 ± 1 and 13.8 ± 1.7 μm in diameter for the Jurkat cells with or without magnetic beads binding, respectively) (*p* > 0.05).

Apart from the fundamental investigation of the maximum velocities of the light images (100% color brightness) that can manipulate the cells, the similar evaluations were also carried out using the light images with varied percentages (i.e., 60, 70, 80, 90, and 100%) of color brightness. Figure 4a–c exhibited the relationship between the maximum velocities of light images that can manipulate the magnetic microbeads-bound Jurkat, OECM1, and HA22T cells, respectively, under varied percentages of color brightness. The results demonstrated that the ODEP force generated on a cell was influenced by the color brightness of light images (Figure 4a–c; *p* < 0.05). The overall trend showed that the ODEP force generated on a cell increased with the percentage increase in color brightness of light images, although some data points revealed that the increase was not statistically significant. As a whole, this finding is line with the result revealed in our previous study [38,39]. 

### 3.3. Design of the Light Image Arrays in the ODEP Microfluidic System for the Purification and Sorting of Cancer Cells with Two Different Sizes

After fundamentally understanding the ODEP operation conditions of the manipulated cells under varied percentages of color brightness (Figure 3 and Figure 4), the light image arrays including two dynamic light image arrays and two static light bar arrays were designed for the purification and sorting of cancer cells (i.e., OECM1 and HA22T cells) with different sizes and for the individual collection of the separated cancer cells, respectively, as illustrated in Figure 2. For the two dynamic light image arrays, the dynamic light image array I with multiple light bars (100% color brightness) was designed to dynamically pool the cells, continuously transported to the cell separation zone, to one side of the microchannel (Figure 2b,c). After that, the dynamic light image array II, encompassing multiple light bars with two different (i.e., 100 and 80%) color brightnesses, was designed to first sort and separate the cancer cells (i.e., OECM1 and HA22T cells) from the magnetic microbeads-bound Jurkat cells and was followed by sorting and separating the OECM1 and HA22T cancer cells by their sizes, respectively. To achieve this goal, the moving velocity of the dynamic light bars of the array I was set at 70 μm s^−1^, which was much lower than the maximum velocities (i.e., 118 ± 21.2, 138.8 ± 16.5, and 184 ± 20.3 μm s^−1^; Figure 3b) of the light bar images (100% color brightness) that can manipulate the magnetic microbeads-bound Jurkat cells and the OECM1 and HA22T cancer cells, respectively. This design ensured that the three types of cells tested can be effectively manipulated and then gathered to the side of the microchannel. The cells manipulated to the side of the microchannel were then released by the three light bars near the side of the microchannel in the flow direction owing to the gradual reduction in light intensities at the end of three light bar images (please also refer to Figure 2 for illustration). In this study, the cell suspension sample was delivered in the microchannel at the set flow rate of 0.4 μL s^−1^.

After being released by the dynamic light image array I, the cells further flowed to the zone of dynamic light image array II, by which they were sorted and separated. In this work, the moving velocity of the dynamic light image array II was set at 130 μm s^−1^, which was higher than the maximum velocities (i.e., 118 ± 21.2 μm s^−1^; Figure 3b) of the light bar images that can manipulate the magnetic microbeads-bound Jurkat cells. Under this circumstance, the magnetic microbeads-bound Jurkat cells could not be attracted and manipulated by ODEP and were thus released by the light bars (100% color brightness) near the side of the microchannel in the flow direction, as schematically illustrated in Figure 2d–f. Conversely, the cancer cells (i.e., the OECM-1 and HA22T cancer cells) were manipulated by ODEP due to the fact that the maximum velocities (i.e., 138.8 ± 16.5, 184 ± 20.3 μm s^−1^, respectively; Figure 3) of the light bar images that can manipulate them were higher than the set moving velocity of the dynamic light image array II (i.e., 130 μm s^−1^). In this situation, the dynamic light bar images (100% color brightness) attracted and pulled the cancer cells to another side of the microchannel (Figure 2d,e). However, when these manipulated cancer cells entered the area where the dynamic light bar images had lower (i.e., 80%) color brightness (Figure 2f), the cancer cells with smaller size (i.e., the OECM1 cancer cells) were not effectively manipulated. This was mainly due to the maximum velocity (i.e., 120.9 ± 20.4 μm s^−1^; Figure 4b) of the dynamic light bar images (80% color brightness) that can manipulate the OECM1 cancer cells being lower than the set moving velocity of the dynamic light images (i.e., 130 μm s^−1^). Therefore, the OECM1 cancer cells were released from the dynamic light image array II and then captured by the static light bar array II (i.e., the three static light bars with 100% color brightness), as illustrated in Figure 2f,g. For the cancer cells with larger size (i.e., HA22T cancer cells), conversely, they were manipulated by the dynamic light bar images with 80% color brightness because the maximum velocity (i.e., 137.1 ± 17.8 μm s^−1^; Figure 4c) of such light bar images that can manipulate the HA22T cancer cells was higher than the set moving velocity (i.e., 130 μm s^−1^) of the dynamic light image array II. As a result, the HA22T cancer cells were manipulated to another side of the microchannel and finally released from the dynamic light bar images with 80% color brightness due to the effect of fluidic flow, as illustrated in Figure 2f. The released HA22T cancer cells were soon captured and collected by the static light bar array I (i.e., the three static light bars with 80% color brightness) (Figure 2g). Based on the design as abovementioned, the presented approach was capable of first purifying the cancer cells from the magnetic microbeads-bound Jurkat cells, followed by sorting and separating the cancer cells with different sizes. The photograph of the designed light image arrays in the ODEP microfluidic system is shown in Figure 5a (a video clip was provided as the Appendix A). 

### 3.4. Performance Evaluation for the Purification and Sorting of Cancer Cells with Two Different Sizes

After the dynamic and static light image arrays were designed, their function for the manipulation of the three types of cells was then tested. First, the dynamic light image array I was designed to pool all the cells (the majority of them were magnetic microbead-bound Jurkat cells) to one side of the microchannel. Due to the laminar flow pattern in a microchannel, the cells pooled to one side of the microchannel would flow along the side of the microchannel, making room in the microchannel for the following cell sorting and separation operation carried out by the dynamic light image array II, as illustrated in Figure 2. In order to test the abovementioned function and its working capacity, the magnetic microbead-bound Jurkat cell suspension with different concentrations (2.5∼10 × 10^4^ cells 100 μL^−1^) was loaded in the microfluidic chip. The magnetic microbead-bound Jurkat cells transported to the area of dynamic light image array I was then manipulated. The removal rate of magnetic microbead-bound Jurkat cells under different concentrations of cell suspension was then experimentally evaluated. Within the experimental conditions tested, the result (Figure 5b) showed that the dynamic light image array I was able to effectively manipulate the magnetic microbead-bound Jurkat cells to the side of the microchannel with a high removal rate (88.1∼89.8%) under the cell concentration range of 2.5∼5 × 10^4^ cells 100 μL^−1^. When the cell concentration was higher than 10^5^ cells 100 μL^−1^, however, the corresponding removal rate might be significantly affected (e.g., the resulting removal rate: 81.5%) (*p* < 0.05). Based on the evaluation, therefore, the cell concentration used in this work was set as 5 × 10^4^ cells 100 μL^−1^.

After investigating the function and working capacity of the dynamic light image array I, the performance of the dynamic light image array II to sort and separate the cells was then experimentally evaluated. In this work, the magnetic microbead-bound Jurkat, HA22T, and OECM1 cells (concentration: 5 × 10^4^, 5 × 10^3^, and 5 × 10^3^ cells 100 μL^−1^, respectively) were individually loaded into the ODEP microfluidic chip. The loaded cells were then treated with the proposed ODEP manipulation scheme (Figure 2). The distribution of the cells tested was then evaluated. The results (Figure 5c, the left figure) revealed that almost all the magnetic microbead-bound Jurkat cells were not found in the two static light bar arrays. Conversely, most of them (99.2%) were collected in the waste sample. Similar evaluations were performed for the OECM1 and HA22T cancer cells. The results (Figure 5c, the middle figure) exhibited that 86.4% of OECM1 cancer cells were collected by the static light bar array II (7.3% of them were collected by the static light bar array I). In this test, only 6.3% of OECM1 was found in the waste sample. The similar result (Figure 5c, the right figure) showed that 82% of HA22T cancer cells were collected by the static light bar array I (18% of them were collected by the static light bar array II). In this evaluation, none of HA22T cancer cells were found in the waste sample. These individual evaluations, overall, demonstrated that the proposed ODEP cell manipulation scheme (Figure 2) and the operation conditions adopted in this study are promising to effectively purify the cancer cells from the background cells (i.e., the magnetic microbead-bound Jurkat cells in this study) and to further sort and separate cancer cells with different sizes.

After the function tests, as described above, the performance of the ODEP microfluidic system for the purification and sorting of cancer cells with two different sizes was then evaluated. For identifying the model cells tested in the work, the HA22T and OECM1 cancer cells were pre-stained with CellTrace™ Calcein Red-Orange (red dots) or expressed with GFP (green dots), respectively. Moreover, a tested cell suspension sample containing the HA22T, OECM1 cells, and magnetic microbead-bound Jurkat cells (ratio: 1:1:5) mimicking a blood sample treated with the immunomagnetic beads-based cell separation process was prepared. After that, the prepared cell suspension sample was loaded into the ODEP microfluidic system and processed with the ODEP manipulation scheme (Figure 2) (a video clip was provided as the Video S2). After the operation, the cells collected in the two static light bar arrays were observed via fluorescent microscopic imaging to evaluate their cell purity. Figure 6a exhibited the fluorescent microscopic images of the cells collected in the two static light bar arrays. It can be found that most of the cells in the array I and II are HA22T and OECM1 cells, respectively. This result was in line with the goal to be achieved in this study (Figure 2). Moreover, the purities of the cells in the array I (i.e., the HA22T cancer cells) and II (i.e., the OECM1 cancer cells) were evaluated to be 90.1 ± 11.3 and 93.5 ± 10.7%, respectively. Furthermore, the cells in the two static light bar arrays can then be harvested individually via further ODEP-based cell manipulation. In practical operation, the static light arrays II can be removed to release the smaller cancer cells (e.g., OECM1 cancer cells in this study) by eliminating the light-illuminated ODEP force blocking. After the smaller cancer cells flow away and are collected at an exit hole, the removal of static light array I facilitates the release and collection of larger cancer cells (e.g., HA22T cancer cells in this study). Using the differences in cell release time would be beneficial for the subsequent collection of cancer cells with different sizes in a single exit channel. A similar concept to the abovementioned process has also been demonstrated in our previously published paper [38]. These purified and sorted cancer cells are found to be valuable for the subsequent applied or fundamental cancer research works. Overall, this study has demonstrated that the proposed ODEP microfluidic system was capable of (1) purifying the cancer cells from the magnetic microbead-bound cells, and (2) further separating and sorting the cancer cells with different sizes in an effective manner. 

## 4. Conclusions

The analysis of CTCs could pave a promising route to comprehensively understand the CTC properties that might be relevant to cancer status. For this goal, the harvest of high-purity and all possible CTCs with different subtypes from the blood samples of cancer patients is crucially important. In addition, the size of CTCs is reported to be relevant to the specific origin of the tumor. Therefore, the initial sorting and separation of CTCs based on their size difference during the CTC isolation and purification process might facilitate the following CTC analytical work. To realize the goals abovementioned, a two-step CTC isolation and purification protocol was proposed. For the first step, the immunomagnetic beads-based cell separation technique was utilized to deplete the majority of blood cells. After that, an ODEP microfluidic system was developed to (1) purify the label-free CTCs from the remaining magnetic microbeads-bound blood cells, and to (2) further sort and separate the CTCs with different sizes. In this study, the ODEP microfluidic system was designed and fabricated. Moreover, the optimum ODEP operating conditions for effective purification and sorting of cancer cells with two different sizes were explored experimentally. Finally, its performance for the purification and sorting of cancer cells with two different sizes was experimentally assessed. The results demonstrated that the presented ODEP-based cell manipulation scheme was able to effectively purify and sort the cancer cells with two different sizes from a tested cell suspension model in a high-purity (achieved purity for the OECM-1 and HA22T cancer cells: 93.5% and 90.1%, respectively) manner. Overall, this study presented an ODEP microfluidic system for the purification and sorting of cancer cells with different sizes. In addition to the application for CTC purification and sorting, the presented method is also useful in other research areas in which the high-purity and label-free purification and sorting of cells with different sizes is required.

## Figures and Tables

**Figure 1 micromachines-14-02170-f001:**
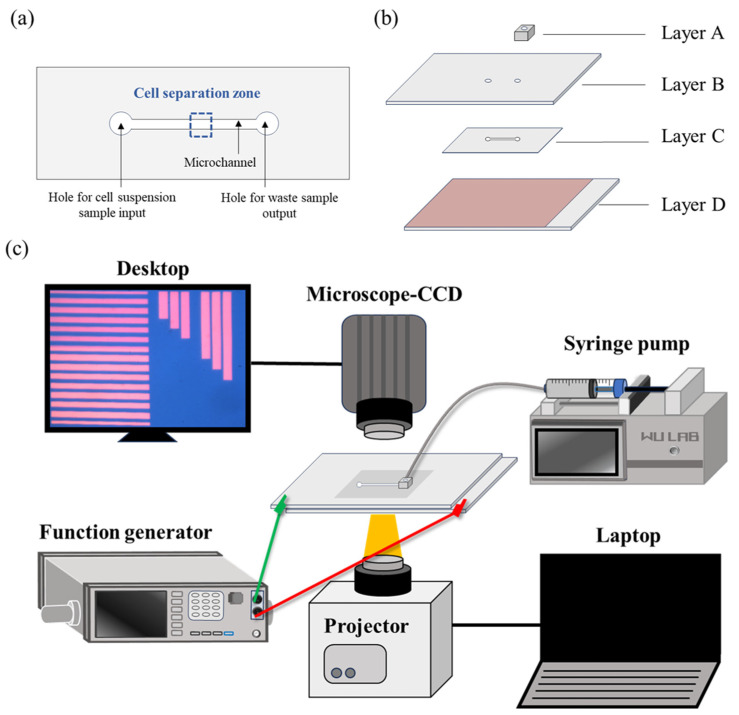
Schematic illustration of the (**a**) layout and (**b**) layer-by-layer structure [Layer A: a processed PDMS substrate; Layer B: an indium-tin-oxide (ITO) glass; Layer C: a processed double-sided adhesive tape with a hollow main microchannel; Layer D: a bottom ITO glass coated with a layer of photoconductive material] of the ODEP microfluidic chip, and (**c**) the overall experimental setup.

**Figure 2 micromachines-14-02170-f002:**
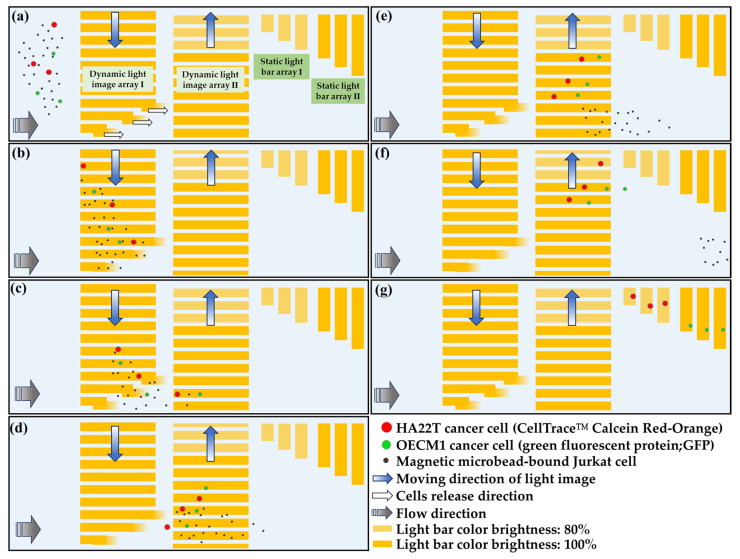
Schematic illustration of the entire ODEP manipulation process for the purification and sorting of cancer cells with different size: (**a**) the designed light image arrays including two dynamic light image arrays and two static light bar arrays (as indicated) in the cell separation zone of the microchannel (the color brightness is as indicated); (**b**,**c**): the dynamic light image array I was used to dynamically pool all cells, continuously transported to the cell separation zone, to one side of the microchannel; (**c**,**d**): the cells manipulated to the side of the microchannel were then released by the three light bars near the side of the microchannel in the flow direction; (**d**,**e**): the cells flowed to the dynamic light image array II were then sorted and separated by the ODEP manipulation, by which the magnetic microbeads-bound Jurkat cells could not be manipulated by ODEP and were thus released by the light bars near the side of the microchannel in the flow direction; (**f**,**g**): the cancer cells (i.e., the HA22T and OECM1 cancer cells) were manipulated by ODEP and thus were attracted and pulled to the another side of the microchannel. During this process, they were sorted and separated based on their size difference by the designed ODEP mechanism and then were collected by the downstream static light bar array I and II, respectively.

**Figure 3 micromachines-14-02170-f003:**
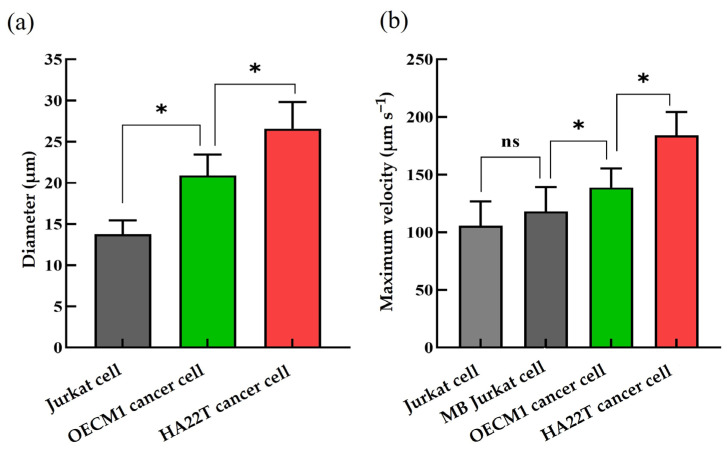
(**a**) The size (diameter) of the model cells (i.e, Jurkat cells and the OECM-1 and HA22T cancer cells) used in this study, and (**b**) evaluation of the maximum velocities of a light image that can manipulate these model cells (i.e., the Jurkat, magnetic microbead-bound Jurkat (MB Jukat), and HA22T, and OECM1 cells). *: significant difference (*p* < 0.05) and ns: not significant.

**Figure 4 micromachines-14-02170-f004:**
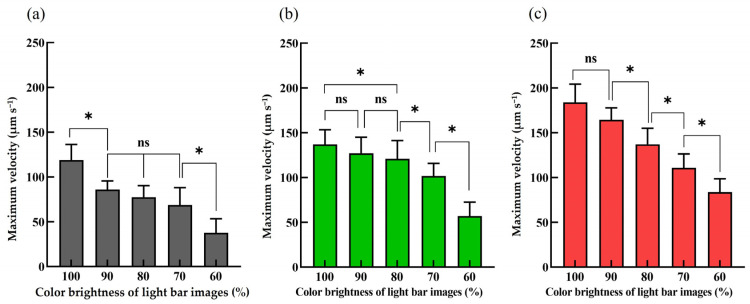
The relationship between the maximum velocities of light images that can manipulate the (**a**) magnetic microbeads-bound Jurkat (MB Jukat), (**b**) OECM1, and (**c**) HA22T cells under varied percentages (60, 70, 80, 90, 100%) of color brightness of light bar images. *: significant difference (*p* < 0.05) and ns: not significant.

**Figure 5 micromachines-14-02170-f005:**
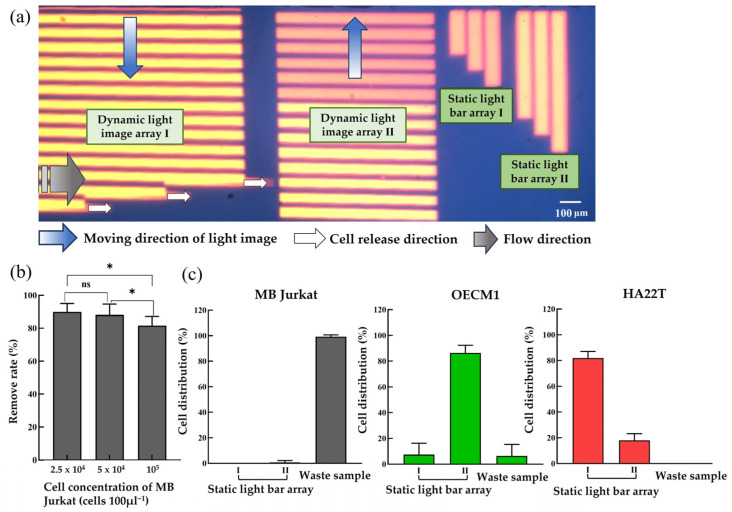
(**a**) The photograph of the designed light image arrays in the ODEP microfluidic system (a video clip was provided as the Appendix A), (**b**) the performance (removal rate %) evaluation of the dynamic light image array I to manipulate the magnetic microbead-bound Jurkat cells to one side of the microchannel under different cell concentration conditions, (**c**) the distribution (%) of the magnetic microbeads-bound Jurkat (MB Jukat) (the left figure), OECM1 (the middle figure), and HA22T (the right figure) cells in the static light bar array I and II and in the waste sample after they were individually treated with the proposed ODEP manipulation scheme, as described in Figure 2. *: significant difference (*p* < 0.05) and ns: not significant.

**Figure 6 micromachines-14-02170-f006:**
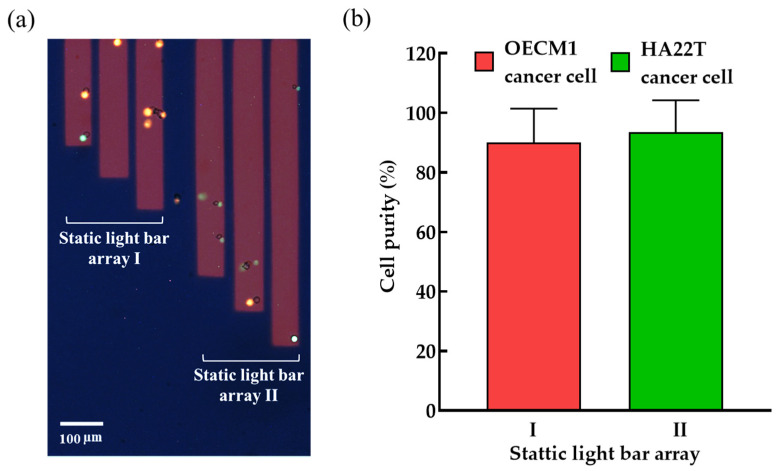
(**a**) The microscopic fluorescent images of the cells collected in the two static light bar arrays (HA22T cancer cells: the red dots; OECM1 cancer cells: the green dots), and (**b**) the evaluated cell purities of the HA22T and OECM1 cancer cells in the static light bar array I and II, respectively.

## Data Availability

Data are contained within the article.

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
