# Peer review of "The Utilization of Optically Induced Dielectrophoresis (ODEP)-Based Cell Manipulation in a Microfluidic System for the Purification and Sorting of Circulating Tumor Cells (CTCs) with Different Sizes"

_micromachines, 2023, doi:10.3390/mi14122170_

Round 1

Reviewer 1 Report

Comments and Suggestions for Authors

In this study, the authors reported the usage of ODEP microfluidic system to separate CTCs with different sizes. The authors have focused onto ODEP-based cell manipulation and separation for more than 10 years. The results shown in this paper are meaningful and convinced, which may lie in high-potential application of high-purity and label-free purification and sorting of cells with different sizes. I think the current form of this paper can be accepted.

Author Response

November 24, 2023

Dear the Reviewer,

We would like to thank the Reviewer for taking the necessary time and effort to review the manuscript. We sincerely appreciate all your valuable comments on our manuscript.

Sincerely,

Min-Hsien Wu, Distinguished Professor
Department of Biomedical Engineering

Chang Gung University

259, Wen-Hwa 1st Road, Kwei-Shan, Taoyuan City, Taiwan, 333, R.O.C.

Reviewer 2 Report

Comments and Suggestions for Authors

The work is well structured and described. The results are clearly illustrated and scientifically sound. Nevertheless some aspects could be improved:
1- in the introduction when the authors say"..Second, the ODEP force generated on a cell is influenced by the intensity (or color brightness ratio) of illuminated light according to our previous study [38,39]." 
There are papers that could be also considered. Among the others Ling, wang-ying et al. Microfluidics and Nanofluidics (2010) and Filippi et al. S&A B (2022). 

 2- please explain which the main novelty of this paper with respect to Yang, Chia-Ming, et al. "Development of an optically induced dielectrophoresis (ODEP) microfluidic system with a virtual gel filtration chromatography (GFC)-inspired mechanism for the high-performance sorting and separation of microparticles based on their size differences." Sensors and Actuators B: Chemical 395 (2023): 134443.

 3- When looking at the lab-on-chip design, some aspects are not clear that the authors should clarify better: Is there only one exit channel? How do they separate the cells attached to the beads from the others? And what about those of two different dimensions?

Comments on the Quality of English Language

please do  revision of the English.

Author Response

Thank you!

Round 2

Reviewer 2 Report

Comments and Suggestions for Authors

the authors satisfied all the reviewer's comments.